# Assessment of *Vibrio parahaemolyticus* levels in oysters (*Crassostrea virginica*) and seawater in Delaware Bay in relation to environmental conditions and the prevalence of molecular markers to identify pathogenic *Vibrio parahaemolyticus* strains

Esam Almuhaideb[1¤a], Lathadevi K. Chintapenta[2¤b], Amanda Abbott[2], Salina Parveen[3], Gulnihal Ozbay[2]*

**1** Department of Human Ecology, Delaware State University, Dover, Delaware, United States of America, **2** Department of Agriculture and Natural Resources, Delaware State University, Dover, Delaware, United States of America, **3** Department of Agriculture, Food and Resource Sciences, University of Maryland Eastern Shore, Princess Anne, Maryland, United States of America

¤a Current address: Department of Agriculture, Food and Resource Sciences, University of Maryland Eastern Shore, Princess Anne, Maryland, United States of America
¤b Current address: Department of Biology, University of Wisconsin River Falls, River Falls, Wisconsin, United States of America

* gozbay@desu.edu

## Abstract

This study identified *Vibrio parahaemolyticus* in oyster and seawater samples collected from Delaware Bay from June through October of 2016. Environmental parameters including water temperature, salinity, dissolved oxygen, pH, and chlorophyll *a* were measured per sampling event. Oysters homogenate and seawater samples were 10-fold serially diluted and directly plated on CHROMagar™ Vibrio medium. Presumptive *V. parahaemolyticus* colonies were counted and at least 20% of these colonies were selected for molecular charcterization. *V. parahaemolyticus* isolates (n = 165) were screened for the presence of the species-specific thermolabile hemolysin (*tlh*) gene, the pathogenic thermostable direct hemolysin (*tdh*)/ thermostable related hemolysin (*trh*) genes, the regulatory transmembrane DNA-binding gene (*toxR*), and *V. parahaemolyticus* metalloprotease (*vpm*) gene using a conventional PCR. The highest mean levels of the presumptive *V. parahaemolyticus* were $9.63 \times 10^3$ CFU/g and $1.85 \times 10^3$ CFU/mL in the oyster and seawater samples, respectively, during the month of July. *V. parahaemolyticus* levels in oyster and seawater samples were significantly positively correlated with water temperature. Of the 165 isolates, 137 (83%), 110 (66.7%), and 108 (65%) were *tlh*+, *vpm*+, and *toxR*+, respectively. Among the *V. parahaemolyticus* (*tlh*+) isolates, 7 (5.1%) and 15 (10.9%) were *tdh*+ and *trh*+, respectively, and 24 (17.5%), only oyster isolates, were positive for both genes. Potential pathogenic strains that possessed *tdh* and/or *trh* were notably higher in oyster (39%) than seawater (15.6%) isolates. The occurrence of total *V. parahaemolyticus* (*tlh*+) was not necessarily proportional to the potential pathogenic *V. parahaemolyticus*. Co-occurrence of the five genetic markers

**Data Availability Statement:** All relevant data are within the paper and its Supporting Information files.

**Funding:** 1- United States Department of Agriculture-National Institute of Food and Agriculture-Capacity Building Grant for Vibrio Project (2014-38821-22430) - funded AA, GO, and SP (https://nifa.usda.gov/). 2- United States Department of Agriculture-Evans-Allen Grant (DELXDSUGO2015 Award) - funded KC (https://nifa.usda.gov/program/agricultural-research-1890-land-grant-institutions). 3- National Science Foundation-Established Program to Stimulate Competitive Research Grant (EPS-1301765), funded GO. The funders had no role in study design, data collection and analysis, decision to publish, or preparation of the manuscript.

**Competing interests:** The authors have declared that no competing interests exist.

were observed only among oyster isolates. The co-occurrence of the gene markers showed a relatedness potential of *tdh* occurrence with *vpm*. We believe exploring the role of *V. parahaemolyticus* metalloprotease and whether it is involved in the toxic activity of the thermostable direct hemolysin (TDH) protein can be of significance. The outcomes of this study will provide some foundation for future studies regarding pathogenic *Vibrio* dynamics in relation to environmental quality.

## Introduction

*Vibrio parahaemolyticus* is a gram-negative, halophilic, pathogenic bacterium that negatively impact aquatic ecosystems and human health [1–3]. They are curved rods, motile with a single polar flagellum and belong to the family *Vibrionaceae*. It is an endemic pathogen in the marine environment that was first identified as a cause of food-borne illness in Japan in 1950 [4, 5]. *V. parahaemolyticus* is one of the key causes of gastroenteritis leading to diarrhea, headache, vomiting, and abdominal cramps following the consumption of contaminated food or water. In addition, this bacterium can cause septicemia and wound infections [3, 6].

In aquatic ecosystems, organisms like oysters which are filter-feeding mollusks, tend to accumulate different microorganisms from seawater during their filtration [7–9]. Therefore, they are able to accumulate *V. parahaemolyticus* 100-fold higher than the surrounding water [8, 9]. During the warmer months, *V. parahaemolyticus* occurrence in oysters can reach 100% [8].

While some *V. parahaemolyticus* strains are associated with marine animal diseases [10], most strains are investigated as a major concern to human health [11]. *V. parahaemolyticus* infections are associated with the consumption of seafood, particularly raw or undercooked oysters, and accounted for 59.5% of laboratory-confirmed *Vibrionaceae* in the United States [11]. In 2006, a total of 177 *V. parahaemolyticus* infections were reported from New York, Oregon, and Washington states, and the laboratory-confirmed cases were over three-fold higher than the average number in all US states during the same period of 2002–2004 [12]. An outbreak of *V. parahaemolyticus* involving three people was reported in Maryland, August 2012 [13]. A multistate outbreak of 16 gastrointestinal illnesses linked to oysters were reported in 2019, and four of them were associated with *V. parahaemolyticus* [14]. The estimated annual mean cost of foodborne illnesses associated with *V. parahaemolyticus* was over US $40 million [15].

*V. parahaemolyticus* strains possess *tlh* species specific gene, which codes for thermolabile hemolysin (TLH) [16, 17]. The virulence of most clinical *V. parahaemolyticus* isolates are associated with the expression of *tdh* (thermostable direct hemolysin (TDH)) and/or *trh* (TDH-related hemolysin-(TRH)) genes [18–20]. Both *tdh*/*trh* genes are associated with β hemolysis on Wagatsuma blood agar, which is known as the Kanagawa phenomenon, and both have been used as accepted genetic markers for the detection of pathogenic *V. parahaemolyticus* in seafood [21–23]. Although, TDH/TRH proteins are the main pathogenic factors in *V. parahaemolyticus* [24], research also shows that many of the clinical isolates possess neither *tdh* nor *trh* genes indicating the potential presence of other virulence-related factors [25, 26].

*V. parahaemolyticus* harbors a *V. parahaemolyticus* metalloprotease (*vpm*) gene that expresses extracellular zinc metalloprotease and shows sufficient proteolytic activity towards type I collagen [27, 28]. *V. parahaemolyticus* metalloprotease can also degrade host tissue and may promote pathogen invasion [2]. On the other hand, metalloprotease has been investigated

and found to be significant as a virulence factor among *Vibrio* spp. [28]. Metalloprotease was reported to have an important role on the extracellular cleavage and activation process of the *V. cholerae* enterotoxic hemolysin into mature hemolysin [29–32]. Therefore, exploring the prevalence and co-occurrence of *vpm* and *tdh/trh* genes in environmental strains of *V. parahaemolyticus* can be of significance.

The transmembrane DNA-binding protein, ToxR, is a regulatory protein in *V. parahaemolyticus* that is encoded by *toxR* gene. The ToxR protein is strongly associated with the upregulation of the gene encoding the virulence toxin TDH [33]. Genome sequencing of pathogenic *V. parahaemolyticus* revealed another virulence factor called type III secretion systems (T3SS), T3SS1 and T3SS2, by which bacterial proteins (effectors) are injected directly into host cells [34]. An infant rabbit model infected with *V. parahaemolyticus* revealed that T3SS2 is essential for intestinal colonization [35]. In addition, T3SS2 is also considered as a prime virulence factor of *V. parahaemolyticus* enterotoxicity [35–37]. It has been reported that ToxR has no role in the production of T3SS2 [38]; however, a later study identified an uncharacterized component of T3SS2 to be critically regulated by ToxR [39]. Furthermore, *toxR* gene is very important to the bile resistance in the intestine, and the *toxR* mutant strains have significantly lower minimal bactericidal concentration compared to the wild strains [40, 41]. In addition, *toxR* gene is required for stress tolerance and colonization of *V. parahaemolyticus* [40]. On the other hand, similar to the *tlh* gene, *toxR* can be a reliable gene for the detection of *V. parahaemolyticus*, and many studies have used it as *V. parahaemolyticus* species-specific gene marker [42–44]. Studies also indicate that *tlh* and *toxR* genes have a compatible and robust result in terms of reliability and specificity for molecular identification of *V. parahaemolyticus* [16, 45]. Findings and reports from previous literature highlight the important relationships among the *tlh*, *trh*, *tdh*, *vpm* and *toxR* genes in terms of pathogenicity and identification of environmental *V. parahaemolyticus* associated with human infections. Therefore, our study aimed to screen the above-mentioned genetic markers to further illustrate the prevalence and patterns of these genetic markers in environmental strains of *V. parahaemolyticus*.

Delaware Bay is the prime oyster ground on the Atlantic coast providing ecological and commercial resources [46]. *V. parahaemolyticus* outbreaks are one of the leading causes for the closures of commercial shellfish industries on the east coast of the United States [47, 48]. This study was conducted to detect and determine total and potential pathogenic *V. parahaemolyticus* levels in oyster and seawater samples from Delaware Bay. Direct plating on CHROMagar Vibrio was used since it is a well-established method, allowing *V. parahaemolyticus* to be simultaneously isolated and differentiated from other *Vibrio* species, and it has a less inhibitory effect on *V. parahaemolyticus* growth than TCBS media [49–51]. This study also examined the correlation of *V. parahaemolyticus* levels in oyster and seawater samples in relation to the physico-chemical parameters. Along with the above-mentioned aims, we reported the co-occurrences of the five genetic markers (*tlh*, *tdh*, *trh*, *toxR*, and *vpm*) in the environmental strains of *V. parahaemolyticus*. To the best of our knowledge there are no published studies on the prevalence and co-occurrence of these genetic markers among *V. parahaemolyticus* in the Mid-Atlantic region. Furthermore, the regional variation in ecology of *V. parahaemolyticus* indicates the need of site-specific data, and this study provides a new set of data specific for the Delaware Bay region.

## Materials and methods

### Study location and sampling

Field sampling collection was granted by the Department of Natural Resources and Environmental Control (DNREC) in 2016 of sampling year. Oysters and seawater samples were

collected once a month from June to October 2016 from Bowers Beach (BB) [39˚03'25.5"N 75˚23'56.8"W] and Lewes, Broadkill (LW) [38˚47'26.3"N 75˚09'50.2"W] in the Delaware Bay. A third site, Slaughter Beach (SL) [38˚56'50.1"N 75˚18'52.4"W] was added to this study from August to October 2016 (Fig 1) to assess a wider range of the Delaware Bay area. Ten to twelve oysters from each site were harvested (one site per week) into Ziploc bags (SC Johnson & Sons, Racine, WI, USA), sub-divided into three groups for biological triplicates (A, B, and C), and placed in an insulated cooler with ice packs to maintain the temperature between 2–10˚C [52]. One liter of seawater was collected from each site at the same time. Water quality parameters such as water temperature, salinity, turbidity, dissolved oxygen, pH, and chlorophyll *a* were recorded onsite using YSI 556 Handheld Multiparameter Instrument (YSI Incorporated, Yellow Springs, OH, USA) to assess the relationship between these parameters and the Colony Forming Units (CFUs) of *Vibrio parahaemolyticus*.

## Processing of oyster and seawater samples

Ten to twelve oysters were collected from each site and divided into three groups to be analyzed in triplicates. For each replicate 3–4 oysters were cleaned upon arrival at the Aquatic Laboratory using a scrub brush and tap water before they were shucked with sterile knives. Oysters tissues and liquors from each replicate were placed into a sterilized blender jar (Waring Commercial, 7010S) and blended for 90 sec at high speed. Twenty-five grams of the blended tissue was diluted with 225 mL of 0.1% Peptone Water (PW; 1 g of peptone [BD, Bacto™ Peptone, 211677], 10 g of NaCl [Fisher scientific, S271], 1 liter of dH2O, pH 7.4 ± 0.2) and blended again for 60 sec at high speed to prepare the homogenate. This homogenate was labelled as the first ($10^{-1}$) dilution. The oyster homogenate and seawater samples from each site were aseptically serial diluted in 0.1% PW to a final dilution of ($10^{-6}$). Following the American Public Health Association Standard [53], one hundred microliters of each dilution [$10^{-1}$ – $10^{-6}$] of both seawater and oyster homogenate samples from each site were aseptically spread plated in duplicate on CHROMagar medium (CHROMagar™ Vibrio, VB912), and incubated for 24 h at 37˚C.

## Identification and isolation of *V. parahaemolyticus*

*V. parahaemolyticus* were identified as mauve colonies on the CHROMagar plates. Each plate with a countable range of 20 to 200 colonies was selected to calculate the number of colony forming units (CFU) of the presumptive *V. parahaemolyticus* [51]. Using a sterile loop, at least 20% of the mauve colonies from each plate were chosen and inoculated aseptically into a 1.5 mL microcentrifuge tube of Tryptone Soy Broth (TSB; Thermo Fisher Scientific Inc, OXOID, CM0129) supplemented with 1% NaCl, and incubated with shaking (175 rpm) overnight at 37˚C (New Brunswick Scientific I 24 Incubator Shaker Series). Microcentrifuge tubes were then centrifuged at 15,000 rpm for 2 min (Eppendorf Centrifuge 5424), and the supernatant was discarded. Equal amounts (600 µL) of Alkaline Peptone Water (APW; 10 g of peptone, 10 g of NaCl, 1 liter of dH2O, pH 8.5 ± 0.2) and TSB + [24% glycerol, BP229, Fisher BioReagents™] were added and the pellet was resuspended and then frozen at -20˚C for further molecular analysis. Samples were prepared for PCR by boiling for 10 min, and immediately chilled on ice (2 min) for cell lysis and DNA release.

## Molecular analysis (PCR procedures and conditions)

Presumptive *V. parahaemolyticus* isolates were further typed for the genetic markers *tlh*, *tdh*, *trh*, *toxR*, and *vpm* using five sets of primers previously assessed [16, 54]. The PCR reaction mixture (10 µL) consisted of 1 µL of cell lysate as DNA template, 2 µL (1.5 mM MgCl$_2$) of the

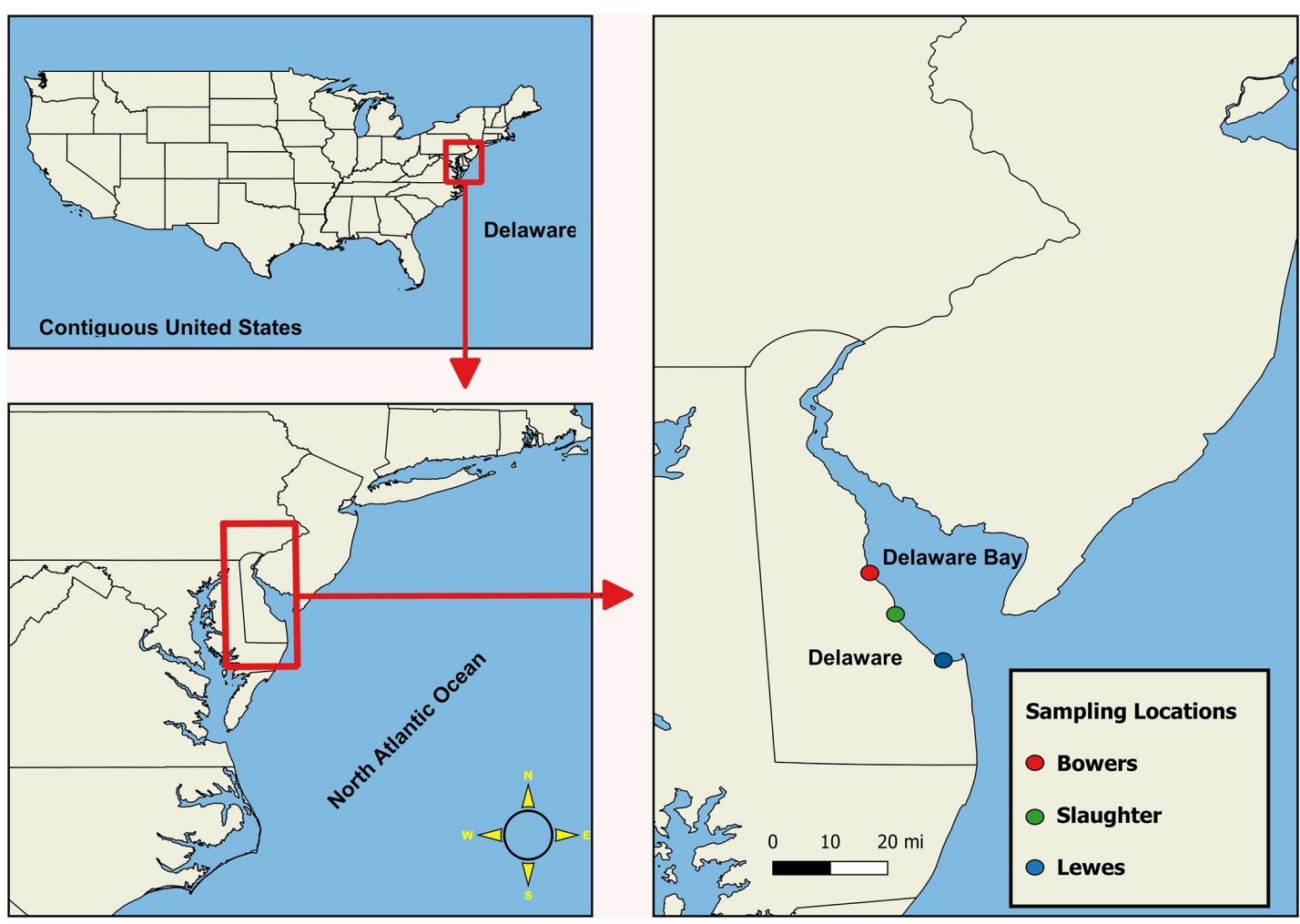

**Fig 1. Study locations in Delaware Bay.**

reaction buffer (5X Green GoTaq® Reaction Buffer; PROMEGA, USA), 0.1 μL (0.5 U) of Taq polymerase (Taq; PROMEGA; USA), 0.4 μL (100 μM) of 2.5 mM deoxynucleotide mix, 0.2 μL (0.2 μM) of each forward and reverse primers (IDT; USA), and 6.1 μL of nuclease free water.

**Table 1. PCR conditions and primers sequences used in this study.**

| Gene | Primer sequences | Cycling conditions |
|---|---|---|
| *tlh* | F-*tlh*: ACTCAACACAAGAAGAGATCGACAA | Cycles: 30 |
| | R-*tlh*: GATGAGCGGTTGATGTCCAA | |
| *tdh* | F-*tdh*: TCCCTTTTCCTGCCCCC | Denaturation temp: 95˚C/30 sec |
| | R-*tdh*: CGCTGCCATTGTATAGTCTTTATC | Annealing temp: 60˚C/45 sec |
| *trh* | F-*trh*: TTGCTTTCAGTTTGCTATTGGCT | Extension temp: 68˚C/1 min |
| | R-*trh*: TGTTTACCGTCATATAGGCGCTT | |
| *toxR* | *toxR*-4: GTCTTCTGACGCAATCGTTG | Cycles: 35 |
| | *toxR*-7: ATACGAGTGGTTGCTGTCATG | Denaturation temp: 94˚C/1 min |
| *vpm* | *vpm* 1: CAGCTACCGAAACAGACGCTA | Annealing temp: 58˚C/1 min |
| | *vpm* 2: TCCTATCGAGGACTCTCTCAAC | Extension temp: 72˚C/1 min |

The amplification conditions for *tlh*, *tdh*, *trh*, *toxR*, and *vpm* genes are shown in (Table 1), and PCR reactions were performed using S1000 thermal cycler (Bio-Rad). One μL of nuclease free water was used for the no template control and 1 μL of *V. parahaemolyticus* SPRC 10290 cell lysate was used as a positive control [55, 56]. Gel electrophoresis (FB-SB-1316; Electrophoresis System; Fisher Scientific; USA) was used to analyze the PCR amplicons in 1% agarose gels containing 0.5 μg/ml ethidium bromide [Fisher BioReagents]. The gels were overlaid with 1x Tris acetate-EDTA buffer and run at 130 V for 30–45 min. DNA bands were visualized using a gel documentation system (Syngene, G: BOX EF).

## Data analysis

For statistical analysis, the CFU values of presumptive *V. parahaemolyticus* were $\log_{10}$ transformed to normalize the data, and the significance level (*P*-value) of 0.05 was used. Spearman's rank correlation analysis was performed to measure the relationship between *V. parahaemolyticus* levels [$\log_{10}$ CFU/g (or mL)] and the parameters affecting water quality (temperature, salinity, dissolved oxygen (DO), pH, turbidity, and chlorophyll *a*). Independent samples t-test was used to determine whether *V. parahaemolyticus* levels [$\log_{10}$ CFU/g (or mL)] among the sample types (oyster and seawater) were significantly different. Statistical analysis was performed using IBM SPSS Statistic software (version 26).

## Results and discussion

### Physico-chemical water quality parameters

Physico-chemical water quality parameters (Table 2) showed that water temperatures ranged from 14.63˚C (LW, October) to 28˚C (BB, August). Salinity levels were in the range of 5.37 ppt (LW, October) to 32 ppt (SL, August). The lowest and highest ranges for dissolved oxygen (DO) (3.12 to 8.23 mg/L) were recorded during the months of August and October from BB and LW sites, respectively. The minimum pH value of 6.44 (LW) and maximum of 8.82 (BB) was observed during the month of October. In terms of turbidity and chlorophyll *a*, the minimum and maximum levels ranged from 19 to 55.35 NTU/FTU and 0.134 to 1.174 μg/L, respectively. Notably, at the LW site and during the month of October, water quality parameters displayed the lowest range of water temperature (14.63˚C), minimum level of salinity (5.37 ppt), highest range of dissolved oxygen (8.23 mg/L), and minimum pH value of (6.44). The seasonal variation between temperature and dissolved oxygen previously reported in the Chesapeake Bay shows that the median temperature (˚C) is inversely correlated with the dissolved oxygen median (mg/L) [57]. Another study from the same region has also reported the lowest dissolved oxygen level (5.3 mg/L), and the highest temperature (29.4˚C) during the month of August [58]. This shows that temperature is inversely correlated with dissolved oxygen concentrations [59].

### Concentration of *V. parahaemolyticus* in oyster and seawater samples

The highest mean levels of presumptive *V. parahaemolyticus* were $9.63 \times 10^3$ CFU/g in the oyster samples during the month of July from BB site. This was higher than *V. parahaemolyticus* (CFU) levels ($6.0 \times 10^2$ CFU/g) detected by direct plating-colony hybridization procedure in Maryland Chesapeake Bay oysters [58]. According to the United States Food and Drug Administration (FDA) regulations and guidance, *V. parahaemolyticus* levels (Kanagawa positive or negative) in this study did not exceed the safety limits ($\geq 1 \times 10^4$ CFU/g) [60]. Clearly, all presumptive *V. parahaemolyticus* (CFU) levels, agree well with the strong correlations between water temperature and *V. parahaemolyticus* densities that are reported in the

**Table 2. Physico-chemical water quality parameters in relation to study sites and date of collection.**

| Site | Date | Temp˚C | Salinity ppt | Turbidity NTU/FTU | Dissolved Oxygen mg/L | Chlorophyll *a* μg/L | pH |
|------|------|--------|--------------|-------------------|-----------------------|----------------------|-----|
| BB | 06/21/2016 | 24.18 | 20.0 | 29.0 | 6.3 | 1.2 | 8.18 |
|    | 07/19/2016 | 27.74 | 27.0 | 19.0 | 3.9 | 0.7 | 7.88 |
|    | 08/02/2016 | 28.00 | 25.0 | 43.5 | 3.1 | 0.8 | 7.88 |
|    | 09/13/2016 | 23.67 | 26.0 | 45.1 | 4.0 | 1.1 | 8.04 |
|    | 10/17/2016 | 17.91 | 25.8 | 55.1 | 8.1 | 0.2 | 8.82 |
| LW | 06/07/2016 | 22.7 | 23.0 | 29.0 | 3.7 | 0.3 | 7.2 |
|    | 07/06/2016 | 22.98 | 32.0 | 33.0 | 4.3 | 0.4 | 7.84 |
|    | 08/08/2016 | 26.43 | 25.0 | 40.8 | 3.4 | 0.2 | 7.55 |
|    | 09/06/2016 | 21.32 | 24.0 | 39.0 | 3.4 | 0.1 | 7.75 |
|    | 10/10/2016 | 14.63 | 5.40 | 54.8 | 8.2 | 0.5 | 6.44 |
| SL | 08/30/2016 | 26.74 | 32.0 | 55.4 | 4.0 | 0.8 | 8.06 |
|    | 09/26/2016 | 20.82 | 26.5 | 55.1 | 4.7 | 0.3 | 7.31 |
|    | 10/24/2016 | 14.68 | 16.6 | 20.0 | 7.8 | 0.3 | 7.44 |

BB: Bowers Beach; LW: Lewes, Broadkill; SL: Slaughter Beach.

literature [58, 61–63], indicating that *V. parahaemolyticus* levels increases with the rise of temperature and vice versa (Table 3). *V. parahaemolyticus* concentrations from seawater samples were notably lower than oyster samples (Table 3), demonstrating that oysters can concentrate the *Vibrio* species higher than 10-fold compared to the surrounding water [8, 9]. This results are in agreement with studies conducted on the Pacific, Atlantic and Gulf Coasts of the United States [58, 64, 65]. However, Independent samples t-test indicated that there was no statistically significant difference in mean *V. parahaemolyticus* $\log_{10}$ CFU/g (or ml) values between sample types (oyster–seawater), $t(24) = 1.159$, $P = 0.258$ ($P > 0.05$). Seawater samples from LW in July, with the highest range of salinity, had low CFU/mL counts compared to BB during the same month indicating that there are parameters other than temperature that may have affected the growth of *V. parahaemolyticus* [61]. During the month of October, *V. parahaemolyticus* levels were undetectable (<10 CFU/g (or mL)) in both oyster and seawater samples from LW and SL sites. However, *V. parahaemolyticus* concentrations in the oyster and seawater samples at site BB were 1.7×10 and 3.3×10 CFU/g or mL, respectively. Although *V. parahaemolyticus* in oysters and seawater was not detectable in some sampling events, the lowest detectable reading in oysters and seawater in this study was 1.7×10 CFU/g (or mL). Figs 2 and 3 demonstrate the $\log_{10}$ CFU/g (or mL) levels of *V. parahaemolyticus* in relation to collection

**Table 3. Averages of *V. parahaemolyticus* CFU/g (or mL) in relation to sample type, study site, and collection time.**

|  | BB-OY | BB-W | LW-OY | LW-W | SL-OY | SL-W |
|------|-------|------|-------|------|-------|------|
| June | 2017 | 33 | 367 | 83 | - [b] | - [b] |
| July | 9633 | 1100 | 1850 | 167 | - [b] | - [b] |
| Aug | 980 | 617 | 1133 | 117 | 117 | 20 |
| Sep | 417 | <10[a] | <10[a] | 17 | 17 | 33 |
| Oct | 17 | 33 | <10[a] | <10[a] | <10[a] | <10[a] |

OY: Oyster; W: Water; BB: Bowers Beach; LW: Lewes, Broadkill; SL: Slaughter Beach.

[a] (Not detectable)

[b] (No sample collection).

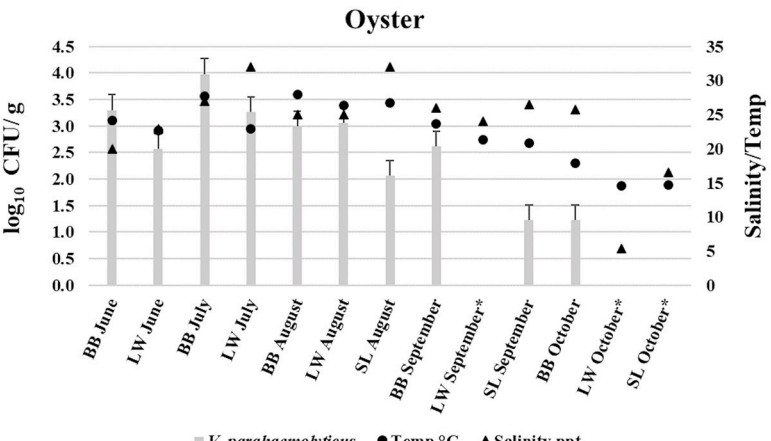

**Fig 2. Average *Vibrio parahaemolyticus* levels (log$_{10}$ CFU/g) in oyster samples in relation to collection time, study sites, and water parameters (temperature and salinity).** Bowers Beach (BB)—Lewes, Broadkill (LW)—Slaughter Beach (SL). * (Not detectable).

time, study site, and water parameters (temperature and salinity) for oyster and seawater samples, respectively.

Spearman's rank correlation analysis showed that water temperature is positively and significantly ($P < 0.05$) correlated to the total *V. parahaemolyticus* (log$_{10}$ CFU/g or mL) levels in oyster and seawater samples (Table 4), which is in agreement with previous studies [58, 61–63]. Salinity had no significant correlation with total *V. parahaemolyticus* (log$_{10}$ CFU/g or mL) levels in oyster and seawater samples (Table 4). Several studies have reported conflicting results regarding the correlations between the abundance of *V. parahaemolyticus* and salinity. Some of these studies have found a correlation between salinity and abundance of *V. parahaemolyticus* [65–68], while others have not [58, 64, 69–72]. Thus, the insignificance of salinity on the abundance of *V. parahaemolyticus* identified in this study cannot be generalized. No significant correlation was found between dissolved oxygen and/or turbidity and the abundance of total *V. parahaemolyticus* (Table 4). This result is in contrast to what has been reported from

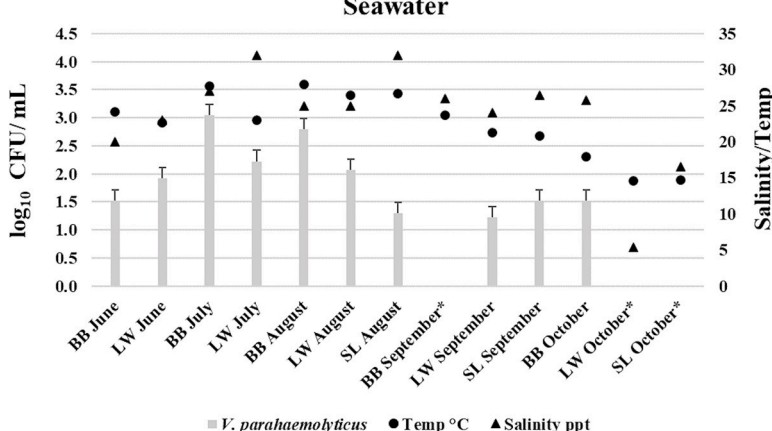

**Fig 3. Average *Vibrio parahaemolyticus* levels (log$_{10}$ CFU/mL) in seawater samples in relation to collection time, study sites, and water parameters (temperature and salinity).** Bowers Beach (BB)—Lewes, Broadkill (LW)—Slaughter Beach (SL). * (Not detectable).

**Table 4. Correlation between *V. parahaemolyticus* log$_{10}$ CFU level and water parameters.**

| Water parameters | Spearman's correlation coefficient, *r* (OY/W)[a] | Oyster (Sig.) | Seawater (Sig.) |
|---|---|---|---|
| Temp | (.777/.639)[b] | (.002)[b] | (.019)[b] |
| Salinity | (.416/.423) | (.157) | (.149) |
| Turbidity | (-.393/-.306) | (.184) | (.309) |
| DO mg/l | (-.368/-.528) | (.216) | (.064) |
| pH | (.416/.142) | (.158) | (.644) |
| Chlorophyll *a* | (.509/.033) | (.076) | (.914) |

[a]OY: oysters; W: seawater.

[b]Correlation is significant at the 0.05 level. Sig. (2-tailed).

studies conducted in the Chesapeake Bay [57, 58]. Both pH and chlorophyll *a* did not significantly correlate with total *V. parahaemolyticus* (log$_{10}$ CFU/g or mL) levels (Table 4), and this is consistent with the previous studies from Mid-Atlantic region [57, 58, 64].

## Molecular identification and characterization of *V. parahaemolyticus*

A total of 165 presumptive *V. parahaemolyticus* isolates (mauve colored on CHROM agar) were further examined for the presence of the species-specific gene (*tlh*), and 137 (83%) were confirmed to be *V. parahaemolyticus* (Table 5). Previous investigation revealed that primers targeting the *tlh*, *toxR*, and *vpm* genes were (100%) specific for *V. parahaemolyticus* strains [16]. The lower occurrence of *vpm* (66.7%) and *toxR* (65.5%) genes compared to *tlh* (83%) gene in this study (Table 6 and Fig 4), suggests that *tlh* gene may occasionally produce false positive results, as a gene similar to *tlh* may occur in other *Vibrio* species specifically *V. alginolyticus* [7, 73]. Yet, regulatory authorities use *tlh* gene as a marker to assess the counts of *V. parahaemolyticus* and reinforce actions to control the outbreaks [74]. This study also showed that (11.7%) of the confirmed *V. parahaemolyticus* possessed only *tlh* gene (Fig 5). In contrast, *toxR* and/or *vpm* genes were only present in coexistence with *tlh*, *tdh*, and/or *trh* (Fig 5), suggest that *toxR* and *vpm* may be more sensitive in detecting *V. parahaemolyticus*. A high percentage of *V. parahaemolyticus* (*tlh*$^+$) were observed among oysters (90.5%) compared to seawater (65.3%) samples (Table 5). At LW and SL sites, *V. parahaemolyticus* were not-detectable (ND) during the month of October (Table 5). About half of the confirmed *V. parahaemolyticus* colonies isolated from seawater (43.8%) possessed only the *tlh* gene indicating the necessity of other gene markers such as *toxR* and *vpm* for *V. parahaemolyticus* strains to survive the internal conditions and colonize oyster gut [2, 40].

Among *V. parahaemolyticus* (*tlh*$^+$), 22.6% and 28.5% were positive for *tdh* and *trh* respectively (Table 6).

This relatively low incidence of *V. parahaemolyticus* (*tdh*$^+$/*trh*$^+$) is in agreement with what has been reported in the literature for environmental isolates [58, 75–77]. Isolates that possessed *tdh*, *trh*, or both *tdh/trh* genes account for 33.5% of *V. parahaemolyticus* (*tlh*$^+$) and were notably higher in oyster (39%) than seawater (15.6%) (Table 6). More than half (61%) of *V. parahaemolyticus* (*tdh*$^+$/*trh*$^+$) isolated from oysters were detected at LW site in July (Table 6). The occurrence of *tdh* and/or *trh* positive *V. parahaemolyticus* was not observed among the study sites during September and October (Table 6). This observation highlights the importance of understanding the dynamics and seasonal variations of pathogenic *V. parahaemolyticus* in Delaware Bay. The high frequency of the *trh* gene (n = 39) compared to the *tdh* gene (n = 31) agrees well with its occurrence in Gulf Coast and Chesapeake Bay oysters [58], and in South Carolina [51]. The co-occurrence of both *tdh/trh* genes was observed in 17.5% and 52%

**Table 5. Occurrence of presumptive and confirmed (*tlh*+) *V. parahaemolyticus* isolates on CHROMagar.**

| Site | Month | Total Presumptive *Vp*. (OY, W) | Total Confirmed *Vp. tlh*⁺ | Confirmed *Vp*. Oyster *tlh*⁺ | Confirmed *Vp*. Water *tlh*⁺ |
|---|---|---|---|---|---|
| BB | June | 16 (14, 2) | 15 (93.8%) | 13 (92.9%) | 2 (100%) |
| | July | 36 (20, 16) | 34 (94.4%) | 20 (100%) | 14 (87.5%) |
| | August | 22 (15, 7) | 18 (81.8%) | 14 (93.3%) | 4 (57.1%) |
| | September | 7 (7, 0) | 7 (100%) | 7 (100%) | 0 |
| | October | 3 (2, 1) | 1 (33.3%) | 1 (50%) | 0 |
| LW | June | 10 (10, 0) | 8 (80%) | 8 (80%) | 0 |
| | July | 40 (33, 7) | 36 (90%) | 31 (93.9%) | 5 (71.4%) |
| | August | 16 (11, 5) | 11 (68.8%) | 7 (63.6%) | 4 (80%) |
| | September | 1 (0, 1) | 0 | 0 | 0 |
| | October | ND | ND | ND | ND |
| SL | August | 12 (3, 9) | 6 (50%) | 3 (100%) | 3 (33.3%) |
| | September | 2 (1, 1) | 1 (50%) | 1 (100%) | 0 |
| | October | ND | ND | ND | ND |
| **Total** | | 165 (116, 49) | 137 (83%) | 105 (90.5%) | 32 (65.3%) |

ND: Not detectable; OY: Oyster; W: Water; BB: Bowers Beach; LW: Lewes, Broadkill; SL: Slaughter Beach.

of total and potential pathogenic *V. parahaemolyticus*, respectively, and they were only among oyster isolates (Table 6). This is in contrast with a study in the Mid-Atlantic [78], in which *V. parahaemolyticus* with both *tdh/trh* genes were observed more frequently among the water isolates. *V. parahaemolyticus* (*tlh*⁺) as illustrated in Table 6 were not proportional to the potential pathogenic *V. parahaemolyticus*, and this agrees well with the previous studies [51, 78].

The occurrence of the five genetic markers showed similar patterns among oyster and seawater isolates (Fig 4). The co-occurrence of the five genetic markers among oyster isolates tested were dominated (43.9%) by *tlh*, *toxR* and *vpm* pattern followed by the coexistence of all five genetic markers (18.9%) (Fig 6). On the other hand, co-occurrence of *tlh*, *toxR*, and *vpm*

**Table 6. Prevalence of *V. parahaemolyticus* genetic markers.**

| Site | Month | *tlh* OY, W | *tdh* OY, W | *trh* OY, W | *tdh/trh* OY, W | *toxR* OY, W | *vpm* OY, W |
|---|---|---|---|---|---|---|---|
| BB | June | 13, 2 | 1, 0 | 5, 1 | 2, 0 | 12, 0 | 11, 0 |
| | July | 20, 14 | 2, 2 | 3, 1 | 1, 0 | 14, 4 | 20, 7 |
| | August | 14, 4 | 0, 0 | 1, 0 | 0, 0 | 14, 1 | 12, 0 |
| | September | 7, ND | 0, ND | 0, ND | 0, ND | 6, ND | 6, ND |
| | October | 1, 0 | 0, 0 | 0, 0 | 0, 0 | 1, 0 | 0, 0 |
| LW | June | 8, 0 | 1, 0 | 0, 0 | 0, 0 | 8, 0 | 7, 0 |
| | July | 31, 5 | 1, 0 | 3, 1 | 21, 0 | 29, 3 | 31, 3 |
| | August | 7, 4 | 0, 0 | 0, 0 | 0, 0 | 7, 2 | 7, 2 |
| | September | ND, 0 | ND, 0 | ND, 0 | ND, 0 | ND, 0 | ND, 0 |
| | October | ND, ND | ND, ND | ND, ND | ND, ND | ND, ND | ND, ND |
| SL | August | 3, 3 | 0, 0 | 0, 0 | 0, 0 | 3, 3 | 2, 2 |
| | September | 1, 0 | 0, 0 | 0, 0 | 0, 0 | 1, 0 | 0, 0 |
| | October | ND, ND | ND, ND | ND, ND | ND, ND | ND, ND | ND, ND |
| Total | | 105, 32 | 5, 2 | 12, 3 | 24, 0 | 95, 13 | 96, 14 |

ND: Not detectable; OY: Oyster; W: Water; BB: Bowers Beach; LW: Lewes, Broadkill; SL: Slaughter Beach.

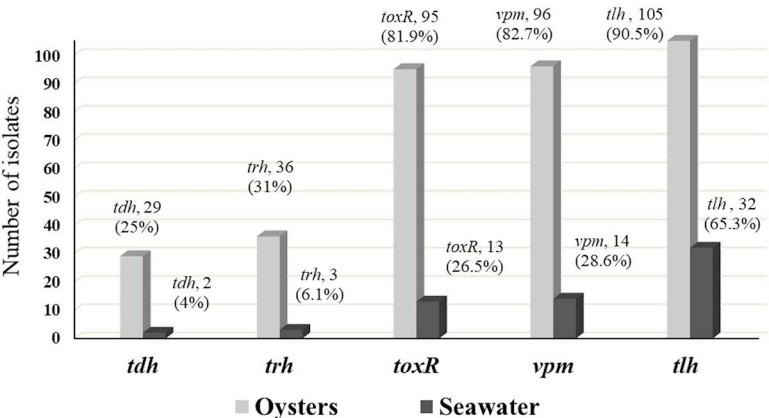

**Fig 4. Percent and number of isolates with detected *tdh*, *trh*, *toxR*, *vpm* and *tlh* genes in oysters and seawater.**

were the second prevalent pattern among seawater isolates (Fig 7). Interestingly and similar to the simultaneous occurrences of *tdh*/*trh*, the coexistence of the five genetic markers were observed only among oyster isolates (Figs 6 and 7), and most of them (19/22) were detected in the LW site where the historical average salinity is close to 26 ppt compared to the BB site with a historical average salinity close to 20 ppt [79]. Variation of gene occurrence patterns among the examined isolates suggest the variation of *V. parahaemolyticus* clones that inhabit Delaware Bay. This study also revealed the relatedness potential of *tdh* occurrence with *vpm*, as Figs 5, 6, and 7 demonstrated that whenever *tdh* was present, *vpm* was also present but not vice versa. This indicates the importance of understanding the role of *V. parahaemolyticus* metalloprotease and whether it is involved in the toxic activity of the thermostable direct hemolysin (TDH) protein as is the case with *V. cholerae* enterotoxic hemolysin [29–32].

## Conclusion

This study assessed *V. parahaemolyticus* levels in oysters and seawater in the Delaware Bay in relation to environmental conditions and the prevalence of key genes. Among the physicochemical parameters assessed in this study, water temperature was the only factor that significantly positively correlated with total *V. parahaemolyticus* level in oyster and seawater samples.

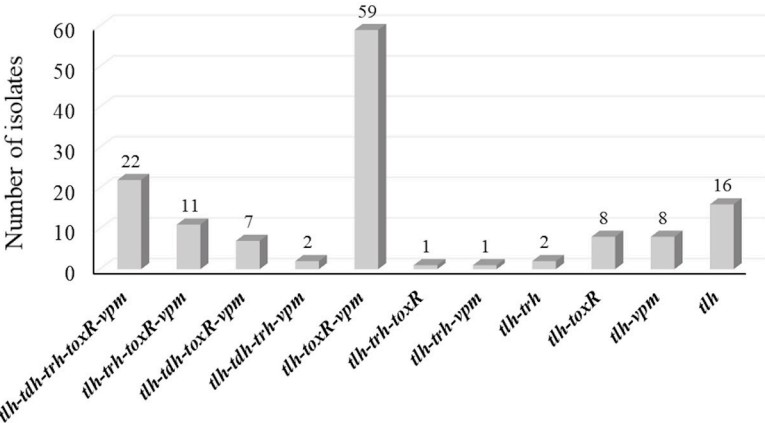

**Fig 5. Number of isolates with coexisting genes among total confirmed *V. parahaemolyticus* isolates.**

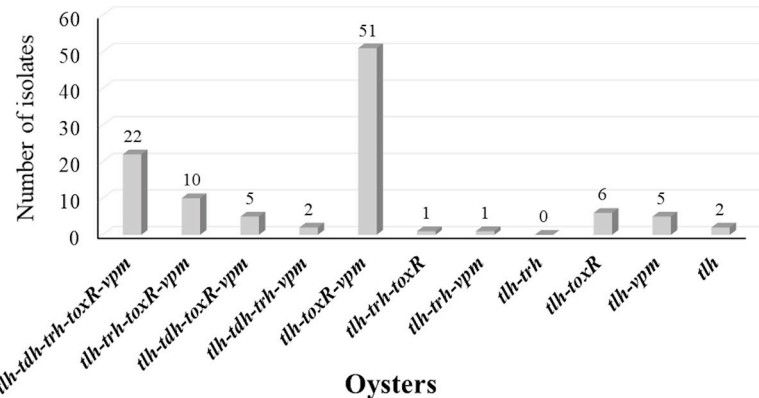

**Fig 6. Co-existence of genes among confirmed *V. parahaemolyticus* isolates from oysters.**

Occurrence of total *V. parahaemolyticus* was not necessarily proportional to the occurrence of potentially pathogenic *V. parahaemolyticus*. The prevalence pattern of the key genes in *V. parahaemolyticus* isolates from seawater does not reflect the pattern of *V. parahaemolyticus* isolates from oysters. The low occurrence of *V. parahaemolyticus* isolates that possessed *tdh*, *trh*, *toxR*, and/or *vpm* genes in seawater samples compared to oysters confirmed the significance of the bioaccumulation process by oysters as a natural nursery for potential pathogenic *V. parahaemolyticus*. Although salinity in this study did not significantly correlate with the *V. parahaemolyticus* level, the historically higher average salinity at Lewes may explain the high frequency of strains from this site that possess all five genes. Utilizing *V. parahaemolyticus* metalloprotease gene (*vpm*) as species-specific gene may provide more accurate results when assessing the prevalence and abundance of pathogenic *V. parahaemolyticus*, and a better understanding of the proportional correlation between the total and potentially pathogenic *V. parahaemolyticus*. The variation among *V. parahaemolyticus* isolates that we have reported indicates the difference in growth rates among Delaware Bay oysters.

Future studies may focus on conducting whole genome sequencing for the *V. parahaemolyticus* isolates to identify the coexistence of the virulence and virulence related genes reported in the literature and illustrate the genetic diversity among *V. parahaemolyticus* isolates inhabiting Delaware Bay. Future studies may also focus on the role of *V. parahaemolyticus* metalloprotease on the toxic activity of TDH. This study provided informative data on oyster-*Vibrio*

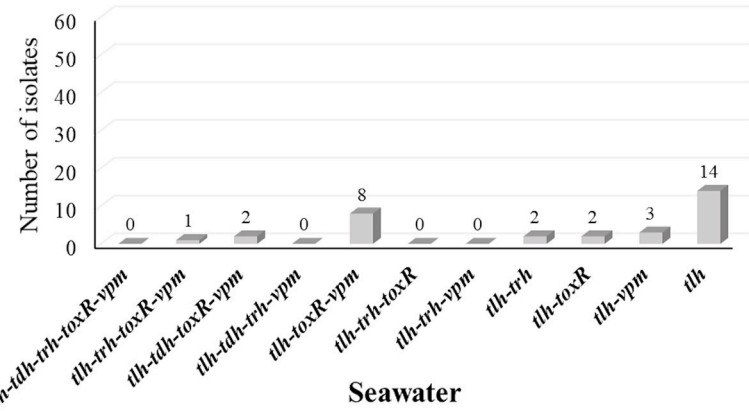

**Fig 7. Co-existence of genes among confirmed *V. parahaemolyticus* isolates from seawater.**

natural contamination factors that can be applied to the risk management programs. The outcomes of this study provide some foundation for future studies regarding pathogenic *Vibrio* dynamics in relation to environmental quality.

## Supporting information

**S1 Table. PCR data for *tlh*, *tdh*, *trh*. *toxR*, and *vpm* genes.**
(XLSX)

## Acknowledgments

We would like to thank Dr. Bettina C. Taylor for valuable comments and editing. We also thank Dr. Gary Richards of USDA ARS for his valuable discussion on our findings.

## Author Contributions

**Conceptualization:** Esam Almuhaideb, Salina Parveen, Gulnihal Ozbay.

**Data curation:** Esam Almuhaideb, Amanda Abbott.

**Formal analysis:** Esam Almuhaideb.

**Funding acquisition:** Lathadevi K. Chintapenta, Amanda Abbott, Salina Parveen, Gulnihal Ozbay.

**Investigation:** Esam Almuhaideb, Amanda Abbott.

**Methodology:** Esam Almuhaideb, Lathadevi K. Chintapenta, Salina Parveen, Gulnihal Ozbay.

**Project administration:** Lathadevi K. Chintapenta, Amanda Abbott, Salina Parveen, Gulnihal Ozbay.

**Resources:** Lathadevi K. Chintapenta, Amanda Abbott, Salina Parveen, Gulnihal Ozbay.

**Supervision:** Lathadevi K. Chintapenta, Gulnihal Ozbay.

**Validation:** Esam Almuhaideb, Lathadevi K. Chintapenta, Amanda Abbott, Salina Parveen, Gulnihal Ozbay.

**Visualization:** Esam Almuhaideb, Lathadevi K. Chintapenta, Amanda Abbott, Salina Parveen, Gulnihal Ozbay.

**Writing – original draft:** Esam Almuhaideb.

**Writing – review & editing:** Lathadevi K. Chintapenta, Amanda Abbott, Salina Parveen, Gulnihal Ozbay.

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
