## [Decision Letter · Decision Letter 0]

31 Jul 2020

PONE-D-20-19028

Analysis of Molecular Markers in the Environmental Strains of Vibrio Parahaemolyticus Isolated from the Oyster (Crassostrea virginica) and Water Samples of Delaware Bay

PLOS ONE

Dear Dr. Ozbay,

Thank you for submitting your manuscript to PLOS ONE. After careful consideration, we feel that it has merit but does not fully meet PLOS ONE’s publication criteria as it currently stands. Therefore, we invite you to submit a revised version of the manuscript that addresses the points raised during the review process.

We look forward to receiving your revised manuscript.

Kind regards,

José A. Fernández Robledo, Ph.D.

Academic Editor

PLOS ONE

Journal Requirements:

2. In your Methods section, please provide additional location information of the study sites, including geographic coordinates for the data set if available.

5. We note that Figure 1 in your submission contain map images which may be copyrighted. All PLOS content is published under the Creative Commons Attribution License (CC BY 4.0), which means that the manuscript, images, and Supporting Information files will be freely available online, and any third party is permitted to access, download, copy, distribute, and use these materials in any way, even commercially, with proper attribution. For these reasons, we cannot publish previously copyrighted maps or satellite images created using proprietary data, such as Google software (Google Maps, Street View, and Earth). For more information, see our copyright guidelines: http://journals.plos.org/plosone/s/licenses-and-copyright.

5.1.    You may seek permission from the original copyright holder of Figure 1 to publish the content specifically under the CC BY 4.0 license.

5.2.    If you are unable to obtain permission from the original copyright holder to publish these figures under the CC BY 4.0 license or if the copyright holder’s requirements are incompatible with the CC BY 4.0 license, please either i) remove the figure or ii) supply a replacement figure that complies with the CC BY 4.0 license. Please check copyright information on all replacement figures and update the figure caption with source information. If applicable, please specify in the figure caption text when a figure is similar but not identical to the original image and is therefore for illustrative purposes only.

Reviewers' comments:

Reviewer's Responses to Questions

**Comments to the Author**

1. Is the manuscript technically sound, and do the data support the conclusions?

Reviewer #1: Yes

Reviewer #2: Yes

2. Has the statistical analysis been performed appropriately and rigorously? 

Reviewer #1: Yes

Reviewer #2: I Don't Know

3. Have the authors made all data underlying the findings in their manuscript fully available?

Reviewer #1: Yes

Reviewer #2: Yes

4. Is the manuscript presented in an intelligible fashion and written in standard English?

Reviewer #1: Yes

Reviewer #2: Yes

5. Review Comments to the Author

Reviewer #1: Dears authors:

The manuscript entitled: "Analysis of Molecular Markers in the Environmental Strains of Vibrio Parahaemolyticus Strains Isolated obtained from the Oyster (Crassostrea virginica) and Water Samples of Delaware Bay", submitted by: Esam Almuhaideb, Lathadevi K. Chintapenta, Amanda Abbott, Salina Parveen Gulnihal Ozbay represents a topic of interest in his field of knowledge, it is well presented and the methodology in general was well conducted; however, before it can be considered for publication, I recommend attending to the following general recommendations:

1) The title requires improvement, it does not specifically reflect the topic it addresses.

2) The abstract does not mention all the methodology used.

3) in the introduction it should be emphasized that molecular markers are relevant to identify pathogenic strains of importance in human health.

4) In the methodology, justify why using the colony count with a presumptive microbiological method and not with a confirmatory method such as coupling the PCR to the most probable number (MPN).

5)En el primer párrafo de la conclusión: "This study assessed V. parahaemolyticus CFU/g (or mL) levels to evaluate the bacteriological conditions of seawater and oyster samples collected from Delaware Bay. V. parahaemolyticus levels in oyster samples were notably higher than seawater samples", se requiere justificar que los métodos de cuantificación son igualmente efectivos para los dos tipos de muestras, sobre todo en cuanto a la cantidad de muestra tomada y analizada.

Likewise, attend to some specific recommendations indicated in the attached document.

Reviewer #2: Analysis of Molecular Markers in the Environmental Strains of Vibrio parahaemolyticus Isolated from the Oyster (Crassostrea virginica) and Water Samples of Delaware Bay

Esam Almuhaideb, Lathadevi K. Chintapenta, Amanda Abbott, Salina Parveen, Gulnihal Ozbay

This study compared levels of Vibrio parahaemolyticus in seawater and oysters in Delaware Bay over 5 months, using culture-based plating as well as genetic markers. Relationships to physico-chemical data were explored using Spearman’s rank correlation analysis, and the co-existence of key genes were assessed.

I believe the manuscript is suitable for publication, but have made some comments below.

I have reattached the manuscript with corrections. Overall, the manuscript needs a grammar polish.

Change sentences so that references are in parentheses and author names not repeated in the sentence

Eg Line 83…

Miyoshi et al. and Kim et al. [24, 25] have reported that V. parahaemolyticus harbors a V.

parahaemolyticus metalloprotease (vpm) gene that expresses extracellular zinc metalloprotease and shows sufficient proteolytic activity towards type I collagen.

Should be written as:

V. parahaemolyticus harbors a V. parahaemolyticus metalloprotease (vpm) gene that expresses extracellular zinc metalloprotease and shows sufficient proteolytic activity towards type I collagen [24, 25]

And

Line 98..

According to Ritchie et al. [32], an infant rabbit model infected with V. parahaemolyticus revealed that T3SS2 is essential for intestinal colonization.

Should be written as:

An infant rabbit model infected with V. parahaemolyticus revealed that T3SS2 is essential for intestinal colonization [32].

Be consistent: U.S. (eg line 118) or USA (line 136)

Line 153: peptone water (PW) – brand or composition?

Line 166: Tryptone Soy Broth (TSB) - brand or composition?

Line 170: Alkaline Peptone Water (APW) - brand or composition?

Line 177: The PCR reaction mixture (10uL) – should be 10 μL

Table 1 is not well laid out. I suggest the following columns:

Gene Primer Sequences Cycling conditions

Line 184: positive control – ATCC or some sort of identification number for V. parahaemolyticus?

Line 201 and elsewhere: 14.63 °C should be 14.63°C (no space between temperature and °C)

Line 202 and elsewhere: space between value and ppt

Line 209: 8.23 mg/L not 8.23mg/L

Fig 2 & Fig 3: It is not accurate to draw the temperature and salinity as lines as they are not a continuous series; either group by location and leave the temperature/salinity as lines, or use a symbol to represent those variables.

Table 4: column headings should read:

Column 1: Water Parameter

Column 2: Spearman’s correlation coefficient, r

Column 3: significance level (oyster)

Column 4: significance level (seawater)

Fig 4 legend should be: Percent and number of isolates with detected tdh, trh, toxR, vpm and tlh genes in oysters and seawater. Y-axis should be “Number of isolates”

Fig 5 legend should be: Number of isolates with coexisting genes among total confirmed V. parahaemolyticus isolates. Y-axis should be “Number of isolates”

Line 328: (n ₌ 39) & (n ₌ 31) should be (n=39) & (n=31)?

Fig 6 legend should be: Co-existence of genes among confirmed V. parahaemolyticus isolates from oysters. Y-axis should be “Number of isolates”

Fig 7 legend should be: Co-existence of genes among confirmed V. parahaemolyticus isolates from seawater. Y-axis should be “Number of isolates”

Conclusion

Line 357-358: Better first sentence needed eg This study assessed V. parahaemolyticus levels in oysters and seawater in Delaware Bay in relation to physico-chemical variables and the presence of key genes.

Line 360 Occurrence of the virulent (should be virulence) genes (tdh and/or trh) in oyster samples were notably higher – higher than what?

Need to mention the relationship of V. parahaemolyticus levels to physico-chemical parameters

Line 371 “virulent” should be virulence

Line 373 “inhibiting” should be inhabiting

References

Check scientific names need to be italicized

I do not have expertise in statistical analyses so cannot comment on the appropriateness of the method used.

6. PLOS authors have the option to publish the peer review history of their article (what does this mean?). If published, this will include your full peer review and any attached files.

Reviewer #1: **Yes: **Josefina León-Félix

Reviewer #2: No

---

## [Author Response · Author response to Decision Letter 0]

7 Oct 2020

Dear Dr. Joerg,

We appreciate that our manuscript is given a chance for publication after revisions. We thank the reviewers for their thorough critique of our manuscript and we have made recommended changes and corrections per reviewers. All the edits and recommendations are incorporated in the manuscript and following section provides our response to those comments and edits:

Reviewer 1

1. The title requires improvement, it does not specifically reflect the topic it addresses.

Changed, please refer to the manuscript. 

2. The abstract does not mention all the methodology used.

All the key methods are included, please see the abstract in the manuscript. 

3. In the introduction it should be emphasized that molecular markers are relevant to identify pathogenic strains of importance in human health. 

We have mentioned the association of the gene markers with clinical V. parahaemolyticus; however, more clarification was added, please see the manuscript lines 82-84, 116-118. 

4. In the methodology, justify why using the colony count with a presumptive microbiological method and not with a confirmatory method such as coupling the PCR to the most probable number (MPN).

1. Direct plating on CHROMagar Vibrio was used since it is a well-established method, less tedious, and sufficient for the detection and isolation of V. parahaemolyticus for research purposes, please see the manuscript lines 125-127, and 171-173.

2. Moreover, in this study, we tested selected colonies (n=165) of the presumptive V. parahaemolyticus isolates for the presence of the species-specific thermolabile hemolysin (tlh) and other key genes of V. parahaemolyticus.

5. In the first paragraph of the conclusion: "This study assessed V. parahaemolyticus CFU / g (or mL) levels to evaluate the bacteriological conditions of seawater and oyster samples collected from Delaware Bay. V. parahaemolyticus levels in oyster samples were notably higher than seawater samples ", it is necessary to justify that the quantification methods are equally effective for the two types of samples, especially in terms of the amount of sample taken and analyzed. 

1. We followed the standard procedure of the American Public Health Association. “Recommended procedures for the examination of sea water and shellfish” in terms of samples collection and inoculation, please see the manuscript lines 165-166.

2. In addition, a few previous studies including our research team reported higher levels of this bacterium in oysters than water. Because oyster as a filter feeder can process large volume of water, please see the manuscript lines 67-68, and 245-248.

6. Likewise, attend to some specific recommendations indicated in the attached document.

Comments in the document:

a) Lines 34-35: The lowest detectable limit of V. parahaemolyticus in oyster and seawater samples was 1.7×10 CFU/g (or mL) during the months of September and October. Confused

Sentence was deleted in the revised abstract because of the space limit since we had to include some information regarding the methods we used, please see the revised abstract of the manuscript.

b) Line 36: Confused. With the temperature of the oysters? It is not documented that the temperature of the oysters or the environment where these oysters are kept has been measured.

Water temperature was measured as is the case with other water parameter, and it is corrected in the revised abstract lines 34-36, 44-45. 

c) Introduction: It would be convenient to make clear that they are focusing on strains of clinical importance or human health, due to the markers that are being evaluated, since there are other pathogenicity markers of relevance in animal health.

Clarification included, please see the manuscripts lines 70-71. 

d) Method: It remains to describe how occurrence, co-occurrence and co-existence were determined.

PCR amplicons were visualized using gel electrophoresis, please see the manuscript lines 195-199.

e) Method: Define enzyme units

Included, please see the manuscript line 189.

f) Method: I recommend placing both added volumes and final concentrations of the PCR components in the reaction

Volumes and final concentrations are included, please see the manuscript lines 187-191.

g) Method: No template control, NTC or blank. Negative control is with unrelated DNA.

Corrected, please see the manuscript lines 193-195.

h) The conclusion is very weak, requires forceful sentences and related to the data obtained.

Please see the revised conclusion. 

Reviewer 2

1. I have reattached the manuscript with corrections. Overall, the manuscript needs a grammar polish. 

All corrected

2. Change sentences so that references are in parentheses and author names not repeated in the sentence

E.g. Line 83…

Miyoshi et al. and Kim et al. [24, 25] have reported that V. parahaemolyticus harbors a V. parahaemolyticus metalloprotease (vpm) gene that expresses extracellular zinc metalloprotease and shows sufficient proteolytic activity towards type I collagen.

Should be written as:

V. parahaemolyticus harbors a V. parahaemolyticus metalloprotease (vpm) gene that expresses extracellular zinc metalloprotease and shows sufficient proteolytic activity towards type I collagen [24, 25]

And Line 98..

According to Ritchie et al. [32], an infant rabbit model infected with V. parahaemolyticus revealed that T3SS2 is essential for intestinal colonization.

Should be written as:

An infant rabbit model infected with V. parahaemolyticus revealed that T3SS2 is essential for intestinal colonization [32].

Corrected here and throughout the manuscript, please see the revised manuscript. 

3. Be consistent: U.S. (eg line 118) or USA (line 136)

USA abbreviation used consistently, please see the revised manuscript. 

4. Line 153: peptone water (PW) – brand or composition?

Composition was included, please see the manuscript lines 161-162.

5. Line 166: Tryptone Soy Broth (TSB) - brand or composition?

The brand name was included, please see the manuscript line 175.

6. Line 170: Alkaline Peptone Water (APW) - brand or composition?

Composition was included, please see the manuscript lines 179-180.

7. Line 177: The PCR reaction mixture (10uL) – should be 10 μL

Corrected, please see the manuscript line 188. 

8. Table 1 is not well laid out. I suggest the following columns:

Gene Primer Sequences Cycling conditions

Changed accordingly 

9. Line 184: positive control – ATCC or some sort of identification number for V. parahaemolyticus?

The positive control was V. parahaemolyticus SPRC 10290, please see the manuscript line 194-195.

10. Line 201 and elsewhere: 14.63 °C should be 14.63°C (no space between temperature and °C)

Corrected here and throughout the revised manuscript.

11. Line 202 and elsewhere: space between value and ppt

Corrected here and throughout the revised manuscript.

12. Line 209: 8.23 mg/L not 8.23mg/L

Corrected, please see the manuscript line 224. 

13. Fig 2 & Fig 3: It is not accurate to draw the temperature and salinity as lines as they are not a continuous series; either group by location and leave the temperature/salinity as lines, or use a symbol to represent those variables.

Lines were changed to symbols, please see the revised figures 2 and 3.

14. Table 4: column headings should read:

Column 1: Water Parameter

Column 2: Spearman’s correlation coefficient, r

Column 3: significance level (oyster)

Column 4: significance level (seawater)

Changed accordingly, please see the revised manuscript.

15. Fig 4 legend should be: Percent and number of isolates with detected tdh, trh, toxR, vpm and tlh genes in oysters and seawater. Y-axis should be “Number of isolates”

Changed accordingly, please see the revised manuscript.

16. Fig 5 legend should be: Number of isolates with coexisting genes among total confirmed V. parahaemolyticus isolates. Y-axis should be “Number of isolates”

Changed accordingly, please see the revised manuscript.

17. Line 328: (n ₌ 39) & (n ₌ 31) should be (n=39) & (n=31)?

Corrected, please see the manuscript line 337. 

18. Fig 6 legend should be: Co-existence of genes among confirmed V. parahaemolyticus isolates from oysters. Y-axis should be “Number of isolates”

Changed accordingly, please see the revised manuscript.

19. Fig 7 legend should be: Co-existence of genes among confirmed V. parahaemolyticus isolates from seawater. Y-axis should be “Number of isolates”

Changed accordingly, please see the revised manuscript.

Conclusion

20. Line 357-358: Better first sentence needed eg This study assessed V. parahaemolyticus levels in oysters and seawater in Delaware Bay in relation to physico-chemical variables and the presence of key genes.

Changed, please refer to the revised conclusion. 

21. Line 360 Occurrence of the virulent (should be virulence) genes (tdh and/or trh) in oyster samples were notably higher – higher than what?

The sentence was deleted in the revised conclusion. 

22. Need to mention the relationship of V. parahaemolyticus levels to physico-chemical parameters

Added please see the revised conclusion. 

23. Line 371 “virulent” should be virulence

Corrected, please see the manuscript line 386.

24. Line 373 “inhibiting” should be inhabiting

Corrected, please see the manuscript line 388. 

References

25. Check scientific names need to be italicized

Checked, please see the revised references.

We thank the reviewers for their careful review of this manuscript. We have made every effort to address all of their comments and revised the manuscript accordingly. We hope that you will now find the manuscript acceptable for publication. 

Sincerely yours,

---

## [Decision Letter · Decision Letter 1]

29 Oct 2020

Assessment of Vibrio parahaemolyticus levels in oysters (Crassostrea virginica) and seawater in Delaware Bay in relation to environmental conditions and the prevalence of molecular markers to identify pathogenic Vibrio parahaemolyticus strains

PONE-D-20-19028R1

Dear Dr. Ozbay,

We’re pleased to inform you that your manuscript has been judged scientifically suitable for publication and will be formally accepted for publication once it meets all outstanding technical requirements.

Kind regards,

José A. Fernández Robledo, Ph.D.

Academic Editor

PLOS ONE

Additional Editor Comments (optional):

Reviewers' comments:

Reviewer's Responses to Questions

**Comments to the Author**

1. If the authors have adequately addressed your comments raised in a previous round of review and you feel that this manuscript is now acceptable for publication, you may indicate that here to bypass the “Comments to the Author” section, enter your conflict of interest statement in the “Confidential to Editor” section, and submit your "Accept" recommendation.

Reviewer #1: All comments have been addressed

Reviewer #2: All comments have been addressed

2. Is the manuscript technically sound, and do the data support the conclusions?

Reviewer #1: Yes

Reviewer #2: Yes

3. Has the statistical analysis been performed appropriately and rigorously? 

Reviewer #1: Yes

Reviewer #2: Yes

4. Have the authors made all data underlying the findings in their manuscript fully available?

Reviewer #1: Yes

Reviewer #2: Yes

5. Is the manuscript presented in an intelligible fashion and written in standard English?

Reviewer #1: Yes

Reviewer #2: Yes

6. Review Comments to the Author

Reviewer #1: (No Response)

Reviewer #2: The authors have addressed the issues raised in the first review. I suggest next time that the manuscript be sent for English editing. Some minor edits are attached.

7. PLOS authors have the option to publish the peer review history of their article (what does this mean?). If published, this will include your full peer review and any attached files.

Reviewer #1: **Yes: **Josefina León Félix

Reviewer #2: No

---

## [Editor Report · Acceptance letter]

20 Nov 2020

PONE-D-20-19028R1 

Assessment of Vibrio parahaemolyticus levels in oysters (Crassostrea virginica) and seawater in Delaware Bay in relation to environmental conditions and the prevalence of molecular markers to identify pathogenic Vibrio parahaemolyticus strains 

Dear Dr. Ozbay:

I'm pleased to inform you that your manuscript has been deemed suitable for publication in PLOS ONE. Congratulations! Your manuscript is now with our production department. 

Kind regards, 

on behalf of

Dr. José A. Fernández Robledo 

Academic Editor

PLOS ONE